# Application of Transfer Effect Models for Predicting Growth and Survival of Genetically Selected Scots Pine Seed Sources in Sweden

**Haleh Hayatgheibi [1,\*], Mats Berlin [1], Matti Haapanen [2], Katri Kärkkäinen [3] and Torgny Persson [4]**

[1] Skogforsk, Uppsala Science Park, SE-75183 Uppsala, Sweden; mats.berlin@skogforsk.se
[2] Natural Resources Institute Finland (Luke), Latokartanonkaari 9, 00790 Helsinki, Finland; matti.haapanen@luke.fi
[3] Natural Resources Institute Finland (Luke), Paavo Havaksentie 3, 90570 Oulu, Finland; katri.karkkainen@luke.fi
[4] Skogforsk, Box 3, SE-91821 Saivar, Sweden; torgny.persson@skogforsk.se
\* Correspondence: haleh.hayatgheibi@skogforsk.se

**Abstract:** We used a regression model approach to examine transferability of the 1.5-generation Swedish Scots pine orchard plus trees using the estimated coefficients of the transfer models recently developed for growth and survival of unimproved Scots pine in Sweden and Finland. Differences between observed and predicted values obtained for height and survival of 3214 plus tree progenies, tested at 58 progeny trials, were regressed on latitudinal transfers ($\Delta$LAT). In order to evaluate rates of improvement in height and survival of selected progenies over unimproved trees, average percentage differences in performances ($\Delta$g%) between the tree groups were calculated. Results indicate that the adopted models can further predict performances of more advanced-generation orchard trees, as there was no evidence of any systematic pattern in the slope of regression functions. Overall, $\Delta$g% estimates obtained for height of progenies were greater than those of survival, suggesting Swedish Scots pine breeding activities are generating gain in the height growth. Moreover, $\Delta$g% estimates obtained for height and survival of half-sib progenies were higher than those of full-sib ones, as a result of response to higher selection intensity applied in the reselection of their parents. This indicates that, in addition to the gain in growth, a gain in survival is also achievable from 1.5-generation seed orchards, depending on the intensity of selection and intended deployment site.

**Keywords:** seed transfer models; seed orchards; genetic gain; scots pine; tree improvement

## 1. Introduction

Plantation of genetically improved reproductive material is of great importance for sustainable forest management. The primary goal in every tree improvement program is to maximize genetic gain in economically important traits. In line with extensive investigations of genetic gain, cited by a recent review [1], breeding programs in Nordic countries have resulted in significant improvements in growth and quality traits [2–4]. The gains from tree breeding are only realized when bred material, such as improved seed, is deployed to the forest [5]. In other words, the principal benefits from any tree improvement program are measured when yield and product quality of improved material are compared with those of plantations established using unimproved material [6].

Seed orchards are by far the most commonly used output system to realize gains from tree breeding, as they deliver the combined effects of selection, testing, and breeding to operational forestry [6,7]. They are often identified by generation, i.e., first-, second-, or advanced-generation orchards [8]. The first-generation orchards are established with grafts obtained from plus trees selected

for their superior phenotypic characteristics in natural stands. These orchards can be upgraded by genetic testing of their progenies across many test locations. This improved orchard variant, composed of genetically-proven selected trees (also called "elite trees"), is often identified as a "1.5-generation" seed orchard [9].

Additionally, seed orchards are used for the production of seed that is well adapted to specific environments [7]. In order to achieve the maximum gain and reduce the risk of maladaptation, seed orchards should preferably be established on the region where the orchard seed is planned to be cultivated [10]. Nevertheless, in cold climates, the amount of seed produced is often too scarce to satisfy the needs of reforestation programs. One practical way to overcome such difficulty is to establish seed orchards in warmer conditions. Such transfers have usually resulted in substantially enhanced seed production [11]. Despite the epigenetic "memory" effect of the female reproductive environment reported in some species [12], the growth rhythm of first-generation founder plus trees is generally similar to that of their original populations from which they were selected [10]. Therefore, experiences from provenance practices, as well as seed transfer models, can be used to develop guidelines for transfer of seed orchard progenies [13].

Scots pine is among the most important and abundant forest tree species native to Eurasia, and has been extensively used in plantation programs in temperate zones [14]. Due to its high commercial importance in Europe, it has been the subject of extensive provenance research and seed-transfer studies since many decades ago [15]. In Sweden, range-wide provenance series were established in the early 1950s [16], from which much valuable information about the genetic variation and effects of transfer on the performance of provenances has been obtained [16,17]. Following this, a seed supply and tree improvement program for Swedish Scots pine was initiated in the beginning of the 1950s, by phenotypic selection (based on growth, straightness, and branching) of 1300 plus trees from different natural stands in Sweden, and establishing the first round of seed orchards [18]. The second round of seed orchards was established in the 1980s, which comprised both progeny-tested plus trees from the initial selection and a second batch of 5500 additionally selected, but still untested, plus trees from young regenerations, also initiated in the 1980s. Approximately a 10% of gain in production at a full rotation was expected from the first round of seed orchards [19]. The third round of seed orchards (equivalent to the 1.5-generation seed orchard) was established between 2004 and 2017, based on the results obtained from progeny testing. The 1.5-generation seed orchards comprise the earlier plus trees reselected based on their superior genetic quality (breeding value) for survival, growth, and branching. As such, in addition to the 23%–27% predicted gain in growth, a gain of 5%–13% in survival is expected from these "elite seed orchards" intended for climatically harsh sites [4,5,20].

Mortality of Scots pine plantations increases in harsh areas [21], thereby increasing both productivity and survival of individual trees are the main objectives for Scots pine breeding in northern Sweden [22,23]. Transfer effect models for growth and survival of Scots pine in Sweden and Finland were recently developed, using unimproved seedlots available in provenance and progeny trials of both countries [13]. It was also proved that the transfer models yielded unbiased predictions of the performance of improved seedlots. The model validation was done for the progenies of first-generation seed orchard plus trees measured in the field and comparing them with their corresponding values predicted in the transfer models [13]. However, the validity of these models for more advanced 1.5-generation orchards remained unknown. Therefore, the main objectives of this study are: (i) to examine the transferability of the 1.5-generation Swedish Scots pine elite seeds using estimated coefficients of the previously developed model functions [13]; and (ii) to compare performances of progenies of phenotypically selected plus trees with those of genetically selected (reselected), and with unimproved trees.

## 2. Materials and Methods

### 2.1. Genetic Field Trials and Measurements

The study material comprises 58 Scots pine progeny trials covering a wide geographic and climatic gradient (61°–67.6°N) in northern Sweden (Figures 1 and 2). The trials are part of the northern Swedish tree improvement program, established by The Forestry Research Institute of Sweden (Skogforsk) between 1971 and 1996, with the main purpose of assessing the field performance of plus tree progenies.

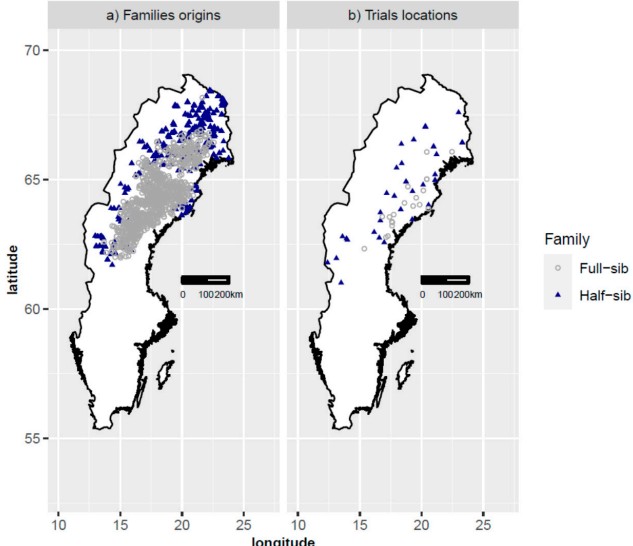

**Figure 1.** Family origins (**a**) and trials locations (**b**). Half-sib and full-sib materials are shown with triangles and circles, respectively.

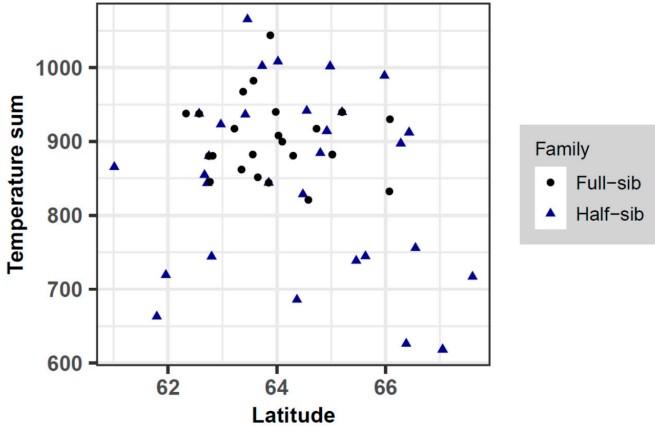

**Figure 2.** Summary of the temperature sum over the latitude of half-sib and full-sib field trials investigated in this study (mid-parents values represent these variables for full-sib families). Half-sib and full-sib families are shown with triangles and circles, respectively.

The trials included 3214 unique plus tree progeny families, divided into 2082 half-sib families in 37 trials and 1132 full-sib families in 25 trials (four trials were in common between full-sib and half-sib families). The 2082 plus-tree parents of the half-sib families were selected as plus trees in 30 to 50-year-old artificially regenerated stands with seedlots of local or unknown origin, whereas the 254 parents of the full-sib families were selected in old autochthonous forest stands.

Of these materials, 141 half-sib and 470 full-sib families representing 141 and 55 elite parent trees, respectively, were reselected and deployed in 1.5-generation seed orchards due to their proven genetic superiority (detailed characteristics of the trials are listed in Table S1). Henceforth, we use the

terms "selected" and "reselected" progenies when referring collectively to the families representing first-generation and 1.5-generation orchard parent trees, respectively.

The environmentally adjusted [13] least-square mean values of height and survival proportion were calculated for each family assessed in each field trial on individual trees at the age of 8–32 years.

The mean values of height and survival of selected families were 380 cm and 0.66, respectively, while for reselected families, they were 460 cm and 0.71, respectively.

Furthermore, natural logarithm and logit transformations were applied to height and survival, respectively, to normalize the data [13].

For the analysis of survival, three of the full-sib trials were removed, as survival of trees had been disturbed due to non-climatic factors, such as moose, fungal diseases, and voles.

## 2.2. Data Analysis and Assessing the Model Performance

In this study, a regression model approach was used to examine transferability of the improved orchard seed sources using the estimated coefficients of the transfer models recently developed for growth and survival of unimproved Scots pine [13]. From now on, we use the term "reference models" when referring to these models [13]. The reaction patterns (i.e., significance of the estimated slope) in plot of regression functions were then used as an indicator of the model performance.

The analyses were conducted in three steps using the R software [24]. Firstly, the expected height and survival were predicted for each family (*i*) in each test environment according to Equations (1) and (2), respectively:

$$\begin{aligned}\hat{Y}_{ij} =\ & -6.0063 + 1.6279\ln(x_1) + 0.156\ln(x_2) + 0.995\ln(x_3) + 0.029(z_1) \\ & -0.1714\left(z_1^2\right) - 0.00005\ (x_3 z_1) + 0.000011\left(x_3 z_1^2\right)\end{aligned} \tag{1}$$

$\hat{Y}_{ij}$ is the log-transformed height of the *i*th family mean in the *j*th trial (*j*), $X_1$ is the age of trees, $X_2$ is the establishment year of the trial subtracted by 1945, $X_3$ is the temperature sum of the trial calculated as degree-days with 5 °C as the threshold temperature [25], and $Z_1$ is the latitudinal transfer (ΔLAT) of the family from the seed-source origin to the location of the trial. ΔLAT was calculated as the difference between the latitude of the origin (the mean latitude of both parents for the full-sib families and the latitude of the mother tree for the half-sib families) and the latitude of the trial.

$$\begin{aligned}\hat{Y}_{ij} =\ & -86.3416 \quad -0.01082(X_1) \\ & +14.2905\ln(X_1) + 0.1626(Z_1) - 0.05642\left(Z_1^2\right) + 0.000864(Z_1 V_1) \\ & -0.00007(Z_1^2 V_1)\end{aligned} \tag{2}$$

$\hat{Y}_{ij}$ is the logit-transformed survival proportion for the *i*th family mean in trial *j*, $X_1$ is the temperature sum of the trial, $Z_1$ is the ΔLAT of the families, and $V_1$ is the altitude of the trial.

Secondly, corresponding transformed values of the observed family means of height and survival were subtracted from their predicted values according to Equation (3):

$$d_{ij} = Y_{ij} - \hat{Y}_{ij} \tag{3}$$

$d_{ij}$ is deviation of the log-transformed height or logit-transformed survival of a family mean ($Y_{ij}$) from the corresponding predicted value ($\hat{Y}_{ij}$) from Equations (1) and (2).

Thirdly, the values of $d_{ij}$ were regressed on ΔLAT:

$$d_{ij} = \beta_0 + \beta_1 z_{ij} + e_{ij} \tag{4}$$

*2.3. Average Performance Differences (Δg%) in Performances between Improved and Unimproved Trees, and Their Comparison among Progeny Groups*

The average percentage differences (Δg%) in height and survival of improved trees, compared to the base level representative of unimproved materials based on which the reference models were developed, were calculated as:

$$\Delta g_{(height)} \; = \; 100 \times \frac{\left( Exp\left( \hat{\bar{Y}}_{ij} + \bar{d}_{ij} \right) - Exp\left( \hat{\bar{Y}}_{ij} \right) \right)}{Exp\left( \hat{\bar{Y}}_{ij} \right)} \tag{5}$$

$$\Delta g_{(survival)} \; = \; 100 \times \frac{\left( \dfrac{Exp\left( \hat{\bar{Y}}_{ij} + \bar{d}_{ij} \right)}{Exp\left( \hat{\bar{Y}}_{ij} + \bar{d}_{ij} \right) + 1} - \dfrac{Exp\left( \hat{\bar{Y}}_{ij} \right)}{Exp\left( \hat{\bar{Y}}_{ij} \right) + 1} \right)}{\dfrac{Exp\left( \hat{\bar{Y}}_{ij} \right)}{Exp\left( \hat{\bar{Y}}_{ij} \right) + 1}} \tag{6}$$

$\hat{\bar{Y}}_{ij}$ is the arithmetic mean of height or survival from Equations (1) and (2), respectively, and $\bar{d}_{ij}$ is the arithmetic mean of height or survival deviations from Equation (3).

## 3. Results

*3.1. Assessing the Regression Model's Performance*

The slope ($\beta_1$) estimates of the log-transformed height deviations regressed on the ΔLAT were non-significant (at a 5% significance level) for both selected and reselected full-sib and half-sib families. However, such estimates were significant for survival of full-sib families, while they were nonsignificant for survival of half-sib families (Table 1). The estimates of intercept ($\beta_0$) for height of plus tree progenies were significantly higher than the baseline. For survival, ($\beta_0$) estimates of half-sib families were significantly higher than the baseline, while they were nonsignificant and significantly lower for reselected and selected full-sib families, respectively (Table 1).

**Table 1.** Estimates of the coefficients obtained for the intercept ($\beta_0$) and slope ($\beta_1$) of the height and survival regression models.

| Trait | Family | Progeny Group | Coefficient | Estimate | *p*-Value |
|---|---|---|---|---|---|
| Height | Halfsib | selected | $\beta_0$ | 0.12 | <0.001 |
| | | | $\beta_1$ | 0.002 | 0.497 |
| | | reselected | $\beta_0$ | 0.17 | <0.001 |
| | | | $\beta_1$ | 0.01 | 0.283 |
| | Fullsib | selected | $\beta_0$ | 0.046 | <0.001 |
| | | | $\beta_1$ | −0.0005 | 0.932 |
| | | reselected | $\beta_0$ | 0.12 | <0.001 |
| | | | $\beta_1$ | 0.013 | 0.058 |
| Survival | Halfsib | selected | $\beta_0$ | 0.082 | <0.01 |
| | | | $\beta_1$ | −0.0023 | 0.91 |
| | | reselected | $\beta_0$ | 0.49 | <0.001 |
| | | | $\beta_1$ | −0.054 | 0.351 |
| | Fullsib | selected | $\beta_0$ | −0.162 | <0.001 |
| | | | $\beta_1$ | 0.077 | 0.0162 |
| | | reselected | $\beta_0$ | 0.032 | 0.32 |
| | | | $\beta_1$ | 0.100 | 0.0279 |

### 3.2. Average Differences (Δg%) in Performances between Improved and Unimproved Trees, and Their Comparison among Progeny Groups

The average differences (Δg%) in performance, in terms of log-transformed height and logit-transferred survival, among different progeny groups, are shown in Table 2. Overall, Δg% estimates of height were higher than those obtained for survival (Table 2). Half-sib and full-sib families reselected for height showed higher Δg% than those of selected families. Additionally, the Δg% estimates obtained for survival of half-sib families were higher than those obtained for survival of full-sib families, particularly for reselected materials (Table 2).

**Table 2.** Average differences (Δg%) in log-transformed growth and survival for full-sib and half-sib families originating from two different progeny groups.

| Trait | Family | Progeny Group | Δg (%) |
|---|---|---|---|
| Height | Halfsib | selected | 12.72 |
| | Halfsib | reselected | 20.05 |
| | Fullsib | selected | 4.75 |
| | Fullsib | reselected | 13.40 |
| Survival | Halfsib | selected | 2.31 |
| | Halfsib | reselected | 11.59 |
| | Fullsib | selected | −3.15 |
| | Fullsib | reselected | 1.42 |

## 4. Discussion

### 4.1. The Model Performance

With advancing tree breeding programs, deployment recommendations based on provenance trials need to be revisited. The reference models previously developed for growth and survival of Scots pine in Sweden and Finland were based on unimproved genetic materials (i.e., provenances and stand-seed check-lots). In our study, the performances of these models, when applied to genetically improved materials, were further tested, as there were some overlaps between the plus tree progeny trials and the trials used for developing the models. Our results revealed that these models can even predict performance of the 1.5-generation elite seeds, which have undergone an intensive genetic selection.

The model residuals were regressed on the latitudinal transfer distance. This approach was adopted because earlier investigations reported that the latitude of the seed source (transfer effect) and the temperature climate (temperature sum) of the site are the main factors influencing the growth and survival of natural-stand seedlings of Scots pine in northern Sweden [25–27]. Correspondingly, reductions in growth and survival rate were commonly reported at sites with a low temperature sum [28].

The residual plots of height and survival showed no systematic patterns, implying the goodness of fit of the reference models. However, the estimated slopes for survival of full-sib families were significant and positive (Table 1). This suggests that when transferred to the south, the improved trees survive better than the model predicts, although this difference is of small practical significance. This finding is consistent with the result of another study which found an increase of the survival rate of full-sib progenies from first-generation Swedish orchard plus trees when planted south of their geographical origin [19]. Overall, all the models developed for transfer effects of Scots pine in Sweden indicate that survival increases with transfer to the south and decreases with transfer to the north [13,29]. However, although the estimated slope was nonsignificant, half-sib trees had a slightly better survival than the model predicted when transferred to the north. When comparing half-sib and full-sib progenies, in terms of their original climatic condition, half-sib families had generally been transferred to harsher sites (i.e., sites at higher latitudes with lower temperature sums) (Figures 1 and 2).

Therefore, high survival of trees under stressful conditions was an important criterion during the reselection of parent trees.

### 4.2. Average Differences in Performances (Δg%) between Improved and Unimproved Trees, and Their Comparison among Progeny Groups

The intercepts of the model functions estimated for height were positive and generally significant (Figure 3, Table 1), suggesting that selection and reselection of plus trees was successful. Contrarily, aside from the north-transferred reselected half-sib progenies, there was no clear superiority in survival over unimproved trees. In a previous investigation, height growth of the full-sib progenies of first-generation Swedish orchard plus trees, measured at ages from 19 to 33 in 36 northern field tests, exceeded those of the unimproved trees by 9.2%, whereas the survival rates were slightly lower than those of the wild-stand check-lots [19].

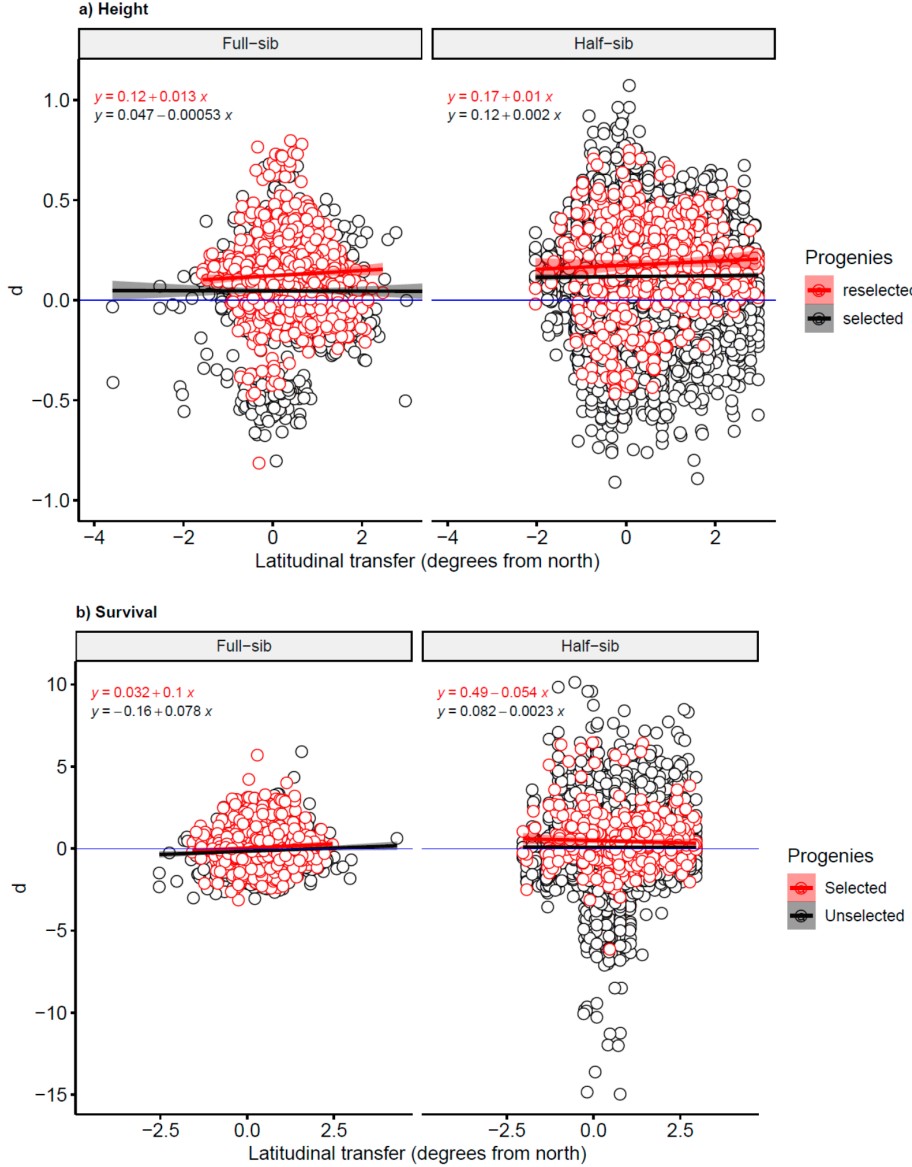

**Figure 3.** The deviation, d, of the recorded measurements from the predicted values for (**a**) log-transformed height and (**b**) logit-transformed survival. On the *X*-axis: negative values = transfers to the north, positive values = transfers to the south.

Rates of improvement in performances of half-sib progenies were much higher than those of full-sib progenies, particularly for survival. This might be a consequence of a higher selection intensity applied in the reselection of plus trees. As a comparison, the half-sib reselected materials were 141 among 2082 selected plus trees (about 7% selection intensity), while the full-sib reselected materials were 55 among 254 selected plus trees (about 22% selection intensity). This finding agrees with the previous report that, in addition to gain in growth, a 5%–13% improvement in survival can be obtained from 1.5-generation orchards, depending on the intensity of selection and deployment site [15].

The first-generation half-sib progenies outperformed their full-sib counterparts in both height growth and survival. This is likely to reflect the way the phenotypic plus-tree selection was performed in these two groups. As noted above, the parent trees of the half-sib families were selected as plus trees in young 30–50-year old artificially regenerated stands [28], whereas the parents of the full-sib families were selected in older natural stands [30]. The results seem to confirm the assumption that phenotypic selection for growth and stem quality is more efficient in even-aged plantations than in mature natural stands where much of the phenotypic among-tree variation is caused by non-genetic factors (ontogenetic and plastic, as well as their interaction).

There are numbers of studies in many tree species from which much extensive information about levels of gain from genetically improved stock has been achieved [1]. However, one limitation of these studies is that the true genetic differences might have been confounded with the long seed transfer effects [31]. Our study is one of only a few in which the transfer effect was eliminated while estimating the difference in performance between improved and unimproved materials. Genetic gain studies require field trials in which improved and unimproved planting stock are compared at similar conditions. Nonetheless, as more than half of the plus-tree progeny trials utilized in this study overlap with the trials used for developing the models, the estimated differences between tree groups in our study are representative and approximately in a similar range (particularly for half-sib progenies) with the genetic gains previously estimated for growth and survival of Swedish Scots pine orchard plus trees.

Comparing half-sib and full-sib progenies in terms of their maximum transfer distances, there are smaller proportions of full-sib progenies transferred to the northernmost and the southernmost sites of their origins, while there are equal amounts of progenies in all tested half-sib trials. This might have an adverse effect on the slope estimates obtained for full-sib progenies. Additionally, the number of half-sib materials investigated was much higher than those of full-sibs, and therefore, we believe that half-sib trials are more representative to provide transfer recommendations in our study.

## 5. Conclusions

This paper represents a study building on 40 years of Scots pine improvement activities in the north of Sweden. Our results indicate that the genetically selected 1.5-generation orchard seed sources can be transferred using the models previously developed for growth and survival of unimproved Scots pine in Sweden and Finland. In general, improved trees were significantly superior in terms of height growth, suggesting Swedish Scots pine breeding activities are generating gain in height. Similarly, genetically selected half-sib progenies had higher performances than those of full-sib progenies, as a result of response to higher selection intensity applied in the reselection of their parents. This indicates that a gain in survival is achievable from 1.5-generation seed orchards, depending on the intensity of selection and intended deployment site.

**Supplementary Materials:** The following are available online at http://www.mdpi.com/1999-4907/11/12/1337/s1, Table S1: Characteristics of the trials.

**Author Contributions:** Conceptualization: T.P., M.B., and H.H.; methodology: H.H. and M.B.; software: H.H.; validation: M.B., M.H., and T.P.; formal analysis: H.H.; investigation: H.H.; data curation: T.P. and M.B.; writing—original draft preparation: H.H.; writing—review and editing: M.H., M.B., T.P., and K.K.; visualization: H.H.; supervision: M.B., T.P., M.H., and K.K.; project administration: T.P.; funding acquisition: T.P., M.B., and K.K. All authors have read and agreed to the published version of the manuscript.

**Funding:** This research was financially supported by The Royal Swedish Academy of Agriculture and Forestry (KSLA) under the "Tandem Forest Values Programme", Norrskogs Forskningsstiftelse, and the European Union Horizon 2020 research and innovation programme under grant agreement No 773383 374 (B4EST project).

**Acknowledgments:** The authors thank Henrik Hallingbäck for discussions during the analysis.

**Conflicts of Interest:** The authors declare no conflict of interest.

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
