# Peer review of "Application of Transfer Effect Models for Predicting Growth and Survival of Genetically Selected Scots Pine Seed Sources in Sweden"

_forests, doi:10.3390/f11121337_

Round 1
Reviewer 1 Report
General comments
In this manuscript the authors address a highly relevant question related to the future of breeding of Scots pine in higher European latitudes, using transfer models based on a wide number of progeny trials.
The paper is rather straightforward in its conception, methods, results and discussion, including a good (and rare) dose of auto-criticism. My only suggestions only require minor (yet, many) changes, due to somehow odd wording here and there.
Specific comments
Lines 33: Please consider changing ‘regeneration stock’ by ‘reproductive materials’.
Line 42: Please delete ‘have been achieved’. Not neccessary
Line 50: Instead of ‘damages’, I think ‘maladaptation’ is a broader term including other aspects of mismatch.
Lines 51: Please use ‘expected’ or ‘planned’ instead of ‘aimed’
Line 52: Pleas be more precise: better ‘scarce’ or ‘not enough to fulfil needs‘ than ‘inadequate’.
Lines 54-55: Not any comment about the well-known problems related to epigenetic embryo memory in Norway spruce?.
Line 59: Please replace ‘which’ by ‘and’.
Line 65: Please replace ‘has been’ by ‘was’.
Line 67: Please replace ‘establishment of’ by ‘establishing’.
Line 70: I think the correct tense is past ‘was expected’…?. Please also check and delete extra spaces throughout the manuscript
Line 74: I understand your meaning by ‘branch quality’ but probably ‘branching pattern’ or simply ‘branching’ is easier to understand to non-breaders.
Line 77: Please clarify the first part of the sentence: plantations, natural stands? And also please, replace ‘productivity and survival’ by ‘increasing both productivity and survival’, or ‘increasing productivity without compromising survival...’
Lines 86-88: Wording of objective ii is too odd. Please rewrite trying a simpler style.
Line 96: Please replace ‘constitute’ by ‘include’
Line 111: Please replace ‘fungi’ by ‘fungal diseases’ or ‘fungal attack’ (remember there are many symbiotic fungi).
Figure 1: Please correct the plot titles: the correct is ‘family origins’ and ‘trial locations’, but better just use a) and b) in the plots, the footer is enough, and it is correct.
Line 120-122: Although absolutely true, this sentence sounds a bit naïve in a scientific journal. Please consider deleting.
Line 127: ‘regression plots’ is clear enough
Line 190: ‘need to’
Lines 192-193: ‘performance was’ or ‘performances were’
Line 207: ‘results’ or ‘the result’
Lines 207-209: ‘…which found an increase of the survival rate of full-sib progenies from first-generation Swedish orchard plus trees when planted south of their geographical origin’.
Line 211: although and half- , please lower case.
Line 215: Please change ‘while they are stressed’ by ‘under stressful environments’ (or conditions)
Line 221: Please change ‘not any clear’ by ‘no clear’
Line 224: ‘Swedish’ here seems unnecessary
Lines 228-229: ‘…reselected materials were 141 among 2082 selected plus trees ( ).... while ...were 55 among 254 ( )?
Line 238: ‘older’ better than ‘elderly’
Line 239-240: ‘is more efficient in even-aged plantations than in mature natural stands’
Line 241: ‘...is caused by non-genetic factors (ontogenetic and plastic and their interaction)’
Line 242: Please delete ‘genetic gain’ (unnecessary)
Line 243: Please replace ‘degrees of gain’ by ‘the levels of gain’.
Lines 246-247: ‘… while estimating the difference in performance between improved and unimproved materials’.
Line 258: ‘to evaluate transfer of trees’ is odd wording...’to provide transfer recommendations’?
Author Response
Many thanks for your great comments. They helped us to improve the manuscript. My replies to the comments were attached. However, after some comments, the reviewers' line numbers miss-matched with the line numbers in the downloaded version.

Reviewer 2 Report
The main aim of the paper is analysing the transfer effect models and its applications in genetically selected scots pines.
The study gives very interesting information about a long-term study in scots pine. The results prove the validity of the model and assess the improvement of the heigh and survival of the trees, however no concrete data is given, only percentage son the real improvement cannot be given. Comparisons are needed with commercial fields to see the real improvement
Specific comments
Line 66. Criteria to improve the trees? Reference given, but information should be given in the introduction
Line 83. Models explanation should be added.
Line 154. Base level is the initial heigh and survival of the trees before selection? How was measure this base level?
Line 185. Table 3 gives gain of the selected and reselected trees, but it is in percentage. As we do not the initial value is a big improvement? Can you give data not about the final height of the selected and reselected trees?
Can you compare the data with commercial stands?
Author Response
Authors are thankful for your comments on the manuscript. responses to your questions were attached.

Round 2
Reviewer 2 Report
The manuscript has been improved, but some of the questions in the previous review remain unclear in the manuscript. The cover letter of the authors solve this questions but they have not included in the manuscript and should be included to improve the paper like height and survival of the trees before and after the selection.
Author Response
Thanks a lot for your comments. We have added the mean values of height and survival of selected and reselected materials used in our study. The mean values of height and survival of unimproved materials (provenances) were further described in Berlin et al. 2016.
However, comparison of performances of progenies using mean values of raw data is a bit biased due to the heterogeneity in trial designs, lack of overlap of genetic entries tested among trial series, large span in establishment years and ages of assessment in each trial. Therefore, the data used in the regression models to compare the percentage differences in performance (gain) of progenies, were subject to a pre-adjustment and standardization resulting in least-square means of survival and tree height per site and genetic entry (which was described in the materials and methods, line 112). Moreover, another importance of our study making it unique compared to other realized gain studies, is that the transfer effect was eliminated while estimating the difference in performances between different groups.
